# Hormetic and transgenerational effects in spotted-wing Drosophila (Diptera: Drosophilidae) in response to three commonly-used insecticides

**Carrie Deans** *[ID]*\*[º], **William D. Hutchison**[º]

Department of Entomology, University of Minnesota, St. Paul, MN, United States of America

[º] These authors contributed equally to this work.
\* dean0179@umn.edu

**Data Availability Statement:** All data are available at the University of Minnesota Data Repository (DRUM) at the following URL: https://hdl.handle.net/11299/226613.

## Abstract

Although insecticide formulations and spray rates are optimized to achieve lethal exposure, there are many factors in agricultural settings that can reduce the effective exposure of insect pests. These include weather patterns, timing of application, chemical degradation/volatilization, plant structural complexity, and resistant populations. While sub-lethal exposure to insecticides can still have negative impacts on pest populations, they can also lead to stimulatory, or hormetic, responses that can increase the fitness of surviving insects. Sub-lethal concentrations may also produce increased tolerance in the offspring of surviving adults through transgenerational effects. Sub-lethal effects are pertinent for the invasive fruit pest, spotted-wing Drosophila, *Drosophila suzukii* (Matsumura), because its small size, diurnal movement patterns, and utilization of hosts with complex plant structures, such as caneberries and blueberries, make effective insecticide applications tenuous. In this study, we measured spotted-wing Drosophila survivorship, reproductive performance, and offspring tolerance in flies exposed to sub-lethal concentrations of three commonly-used insecticides (zeta-cypermethrin, spinetoram, and pyrethrin). We found some evidence for hormesis, with survival effects being sex- and concentration-dependent for all insecticides. Males were far more susceptible to insecticides than females, which in some cases exhibited higher eclosion success and reproductive rates when exposed to sub-lethal doses. We did not observe significant transgenerational effects at sub-lethal concentrations, despite trends of increased offspring viability for zeta-cypermethrin and spinetoram. More research, however, is needed to fully understand the role that sub-lethal effects may play in pest population dynamics, insecticide efficacy, and the development of genetic resistance.

## Introduction

Spotted-wing Drosophila, *Drosophila suzukii* (Matsumura), an invasive species native to Asia, has become a serious pest in fruit-producing areas throughout the United States and Europe,

**Funding:** This work was supported by funding from the Agricultural Growth, Research, and Innovation (AGRI) Crops Research Program, of the Minnesota Department of Agriculture, St. Paul, MN (2018-2021), and the Minnesota Agricultural Experiment Station, University of Minnesota, St. Paul, MN, USA. Grants were awarded to WH. The funders of this study had no role in study design, data collection and analysis, decision to publish, or preparation of the manuscript.

**Competing interests:** The authors have declared that no competing interests exist.

following its establishment in the early 2000s [1–4]. The ability of spotted-wing Drosophila to infest ripening intact fruit has allowed the pest to occupy unique niches in invaded regions, while its ability to utilize non-crop hosts and overwinter has also led to the establishment of stable populations in regions of large-scale fruit production. High population densities, coupled with strict zero-tolerance infestation limits in fresh berry markets, have led to the increased use of broad-spectrum insecticides for control. Many commercial conventional and organic insecticides are available for use on spotted-wing Drosophila, and overall, these products are effective [5–9] but it is not uncommon for growers to perform 4–9 seasonal applications depending on the crop [10–12]. Reductions in efficacy may be due to resistance, which has been detected for some chemistries in California [13], Michigan [14], and Georgia [15]. However, other factors, such as weather [12, 16], timing of application [12, 17–19], pesticide volatilization/degradation [20–22], and penetration into cultivars with high structural complexity [23], plant nutrient content [24–26], and insecticide resistance [14, 27, 28], can also reduce insecticide efficacy. This can occur by impeding contact between the insecticide and spotted-wing Drosophila adults, considering that spotted-wing Drosophila are small, highly mobile, crepuscular insects that are documented to exhibit daily movement patterns [29–33]. It can also occur due to environmental factors that degrade insecticidal compounds or cause run-off, which can reduce the residual activity or effective exposure of insect pests. These factors can cause spotted-wing Drosophila populations to be occasionally or routinely subjected to sub-lethal doses of insecticides, even when growers adhere to the recommended concentrations and spray rates. While sub-lethal exposure to insecticides may still produce negative effects on insect performance [34–37], evidence of stimulatory effects, via hormetic and/or transgenerational effects, have also been documented. Hormetic and transgenerational effects can not only directly increase plant damage through stimulatory effects, they may also promote the evolution of insecticide resistance through their impacts on phenotypic diversity [38–45].

Hormesis occurs when a stressor, such as an insecticide, produces a stimulatory effect at low doses and an inhibitory effect at higher doses. It is typified by a J- or U-shaped dose-response curve [46, 47]. Hormetic responses in insects have been documented since the mid-1940's [48–50], corresponding with the rise in synthetic insecticide use. However, hormetic research didn't increase substantially until the 1980's [51–54]. Hormesis, as measured by increases in longevity [50, 54–56], reproductive rate [51, 52, 57–68], body size [53, 54, 56, 69–72], oviposition [63, 73–77], and growth rate [78–80] have been reported for a wide range of different insect genera and chemical compounds [80]. Despite the wealth of studies documenting hormetic effects at low dose ranges for different insecticides, the implication of these effects for pest control efficacy have been largely under-explored, but see [80]. Though addressed by some [80–89], it is surprising that the effects of hormesis have not been more thoroughly discussed in agricultural pest management, considering that there are a multitude of factors in agricultural settings that may reduce effective exposure and there are significant costs associated with insecticide failures due to increases in insect tolerance or genetic resistance. Another, often-overlooked, impact of sub-lethal exposure is transgenerational tolerance [90, 91]. Transgenerational effects, sometimes referred to as maternal or paternal effects, occur when sub-lethal parental exposure to a toxin leads to increased tolerance in their offspring [40]. In agricultural systems transgenerational effects have been documented for aphids exposed to imidacloprid [81] and thiamethoxam [89] nitenpyram [92], and carbamate [93]. Transgenerational response to insecticides have also been reported for white flies [94] caterpillars [95], and other insect predators [96–98] in agricultural systems. These effects are purported to be mediated by epigenetic mechanisms, such as heritable gene methylation and/or histone modification, or

the transfer of cytoplasmic molecules like RNAs [40, 99, 100]; however, the underlying mechanisms of these response are rarely studied.

While the development of genetic resistance arguably poses a greater threat to the viability of current chemical controls, gene-by-environment interactions associated with sub-lethal exposure, including hormesis and transgenerational effects, may also pose significant threats to insecticide efficacy. This is particularly true given that a multitude of different factors can reduce effective insecticide exposure in agricultural systems, that, when taken together, can lead to cumulative effects that are potentially more widespread. Sub-lethal effects can also generate diverse phenotypes that may serve as adaptive intermediate stages in the evolution of insecticide resistance [40, 101–103]. When taken together, the impact of these issues may rival that of genetic resistance.

In this study, we measured the sub-lethal effects of three commonly used insecticides for spotted-wing Drosophila, including a synthetic pyrethroid, zeta-cypermethrin (Mustang Maxx), a spinetoram (Delegate), and an organic broad-spectrum pyrethrin (Pyganic). More specifically, we aimed to determine whether hormetic and/or transgenerational effects were detectable when spotted-wing Drosophila adults were exposed to sub-lethal concentrations. Spotted-wing Drosophila is a relatively new invasive pest that utilizes a broad range of fruit cultivars that vary in structural complexity [3]. It is also a small, highly mobile pest that displays distinct diurnal movement patterns [30–32]. These characteristics, in addition to other environmental factors, may act as obstacles to achieving lethal practical exposures in the field. Therefore, determining the impact that sub-lethal exposure may have on spotted-wing Drosophila survivorship and/or performance is important for maintaining and/or optimizing insecticide protocols. Furthermore, documenting the prevalence of hormetic and transgenerational effects is needed for further understanding the extent to which gene-by-environment interactions may contribute to the development of insecticide tolerance and/or resistance.

## Materials and methods

### Fly culture

All flies used in the experiments were from a lab colony established in 2018 from infested raspberry samples collected at the UMORE Station in Rosemount, MN. Flies were kept in clear narrow polystyrene vials with foam plugs (Genesee Scientific, San Diego, CA), and each vial contained approximately 5ml of a standard cornmeal-based oligidic diet (cornmeal, sugar, agar, nutritional yeast, propionic acid, methyl paraben, ethanol), as well as a strip of filter paper to reduce condensation. Each vial contained approximately 50 adults, which were transferred to new diet every 2–3 days and kept in a walk-in chamber at ambient lab temperature (20–22˚C) under a 14:10 light-day cycle [104].

### Insecticide exposure

For all insecticide exposures, newly-eclosed flies (24–48 hrs old) were anaesthetized with $CO_2$ and placed in glass scintillation vials that were coated with a specific concentration of insecticide residue. For all insecticide exposures, including the initial dose-response assays, we used Mustang Maxx (FMC Corporation, Philadelphia, PA) as a source of zeta-cypermethrin, Delegate (Corteva Agriscience, Indianapolis, IN) as a source of spinetoram, and Pyganic (MGK, Minneapolis, MN) as a source of pyrethrin. Zeta-cypermethrin formulations used acetone as a solvent, while spinetoram and pyrethrin formulations were water-based. Control flies were placed in vials with no insecticide coating to standardize for $CO_2$ anaesthetization and vial conditions. The exposure period for all experiments was 4 hours, after which flies were transferred to rearing vials containing standard fly diet.

## Dose-response assays

Before each experiment, dose-response assays were conducted for each insecticide. Mortality was recorded after 4 hours of exposure to zeta-cypermethrin (6 concentrations: 0, 0.05, 0.1, 0.5, 1, 20), spinetoram (7 concentrations: 0, 0.01, 0.1, 1, 10, 100, 1000), and pyrethrin (8 concentrations: 0, 0.001, 0.01, 0.1, 1, 10, 50, 140), respectively. Six replicates were used for each concentration and each vial contained 3 male and 3 female flies. The results of these initial assays were used to select the doses used for the sub-lethal and transgenerational treatments and were based on a probit analysis that estimated the concentrations associated with varying degrees of lethality.

## Hormesis protocol

We measured survival and performance across five sub-lethal concentrations of three of the most commonly used insecticides for spotted-wing Drosophila control in the Midwest United States [105]: zeta-cypermethrin, spinetoram, and pyrethrin. We tested concentrations that corresponded to the $LC_0$, $LC_{10}$, $LC_{20}$, $LC_{30}$, $LC_{40}$ values determined by our initial dose-response assays. We tested single-exposure regimes for all three insecticides but also a double-exposure regime for spinetoram and pyrethrin. Table 2 shows the exact concentrations of each insecticide used for each treatment. Flies in the single-exposure treatments were exposed at the start of the experiment, while flies in the double-exposure treatments were exposed at the start of the experiment and again 5 days later. During the treatment exposures, flies were placed in insecticide-treated vials for 4 hours then transferred into new insecticide-free vials with fresh diet. Flies were subsequently transferred to new vials with new diet three times for the rest of their lifespan. Each treatment consisted of 10 replicate vials containing 5 male and 5 female flies. Mortality was recorded daily for 10 days. The total number of pupae and emerging adults, as well as the average adult mass of F1 flies per day, were recorded for each vial throughout the entire lifecycle.

## Transgenerational protocol

To determine if parental exposure had any impact on offspring susceptibility, we exposed adult parental flies (P) to one of 6 sub-lethal concentrations ($LC_0$, $LC_{10}$, $LC_{15}$, $LC_{20}$, $LC_{25}$, $LC_{30}$ doses) of cypermethrin, spinetoram, or pyrethrin and then measured their offspring's (F1) susceptibility to a single diagnostic dose. Newly eclosed male and female P flies were exposed via a 4-hr vial assay then moved into new untreated diet vials containing 4 males and 4 females from the same exposure treatments. They were allowed to mate and lay eggs for 7 days. Each vial was transferred to new diet every 2–3 days for one week. Once enough F1 adults were available (~15 days after P exposure) groups of newly-eclosed 5 males/5 females were exposed to a ~$LC_{25}$ dose of their respective insecticide, except for pyrethrin, which was exposed to a higher dose (~$LC_{50}$) after observing unexpectedly high survivorship across the sub-lethal parental doses (0.1 ppm for zeta-cypermethrin, 1.0 ppm for spinetoram, and 0.1 ppm for pyrethrin. Mortality was recorded after a 4-hour vial exposure period. The mortality of F1 flies from the P control treatment, which was not exposed to insecticide, was compared to the F1 mortality across the other P treatments do determine if their susceptibility was statistically different.

## Data analysis

A probit analysis was used to analyze the dose-response results and determine the lethal concentrations or each mortality rate. For the hormesis survival data, survivorship at the end of 11

days was analyzed in a three-way ANOVA for the spinetoram and pyrethrin datasets, with sex, exposure (single or double), and treatment as main factors. For the zeta-cypermethrin data, only sex, treatment, and sex*treatment were included as factors. The same ANOVA models were used for the performance data, which included the number of pupae (adjusted for the number of living females per vial per day, or pupae/female/day), eclosion success (the proportion of pupae that eclosed as adults), and average adult mass (the mass of all eclosing adults each day divided by the total number of adults). Each insecticide was analyzed separately, as experiments were done at different times. Datasets were rank-transformed when necessary to meet normality assumptions and a Tukey's HSD test (with a Bonferroni correction) was used for post-hoc comparisons. For the transgenerational data, the effects of sex and parental treatment on the mortality of F1 adults were analyzed using an ANOVA, with a Tukey's HSD test for post-hoc analyses (data were rank-transformed to meet normality assumptions). All statistical analyses were performed in SPSS v.27 (IBM Corp, 1989, 2020).

## Results

### Dose-response assays

Table 1 shows the summary statistics for each initial dose-response probit analyses. We achieved good fits for each curve, with all Chi-square analyses being non-significant, and all $R^2$ values being greater than 0.80. Although the susceptibility of the lab colony to the different insecticides did vary, the dose-response assays were performed at different times, so no statistical comparisons were made. Adult flies were initially more susceptible to pyrethrin and less susceptible to spinetoram, with susceptibility to zeta-cypermethrin being intermediate. Fig 1A–1C shows the dose-response curves for each insecticide, which vary in the magnitude of the hormetic responses and dose ranges over which it occurs. Both the hormetic response and dose range for zeta-cypermethrin (Fig 1A) is considerable smaller than that for spinetoram (Fig 1B) and pyrethrin (Fig 1C). Our $LC_{50}$ values for zeta-cypermethrin and spinetoram were comparable to those found for Michigan populations and reported in [9], with the zeta-cypermethrin $LC_{50}$ being slightly higher and the spinetoram $LC_{50}$ being comparable to Michigan populations before increases in insecticide use. Insecticide concentrations corresponding to specific mortality levels based on the initial probit analyses are shown for each insecticide in Table 2.

### Hormetic effects

The hormesis data for zeta-cypermethrin showed significant effects of sex and treatment but no impact of exposure type or any interactions (Table 3). Overall, female mortality was 1.4 times higher than that of male flies. Fig 2A shows that survivorship was highest in the LC10 treatment for males and in the LC20 treatment for females. Despite this, survivorship across the $LC_0$, $LC_{10}$, and $LC_{20}$ concentrations was statistically similar but decreased significantly at the $LC_{30}$ and $LC_{40}$ concentrations (S1 Table). S1A and S1D Fig shows that survivorship for both sexes was high across all treatments until day 8, when mortality sharply increased for the

**Table 1. Statistics for the initial dose-response probit analysis for each insecticide.**

| Insecticide | $X^2$ | df | P-value | slope | intercept | $R^2$ |
|---|---|---|---|---|---|---|
| zeta-cypermethrin | 0.141 | 2 | 0.932 | 0.444 | 4.702 | 0.842 842 |
| spinetoram | 0.231 | 3 | 0.972 | 0.549 | 4.370 | 0.812 |
| pyrethrin | 0.700 | 2 | 0.705 | 0.725 | 5.817 | 0.972 |

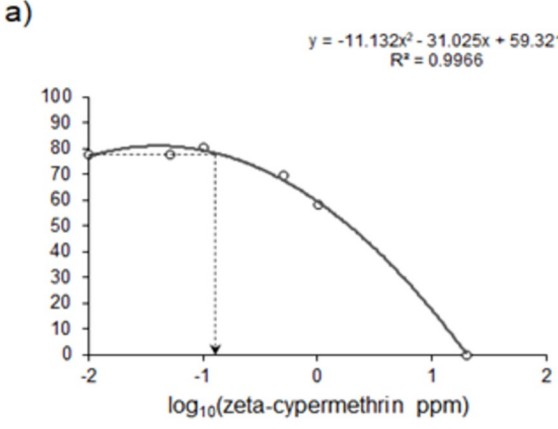

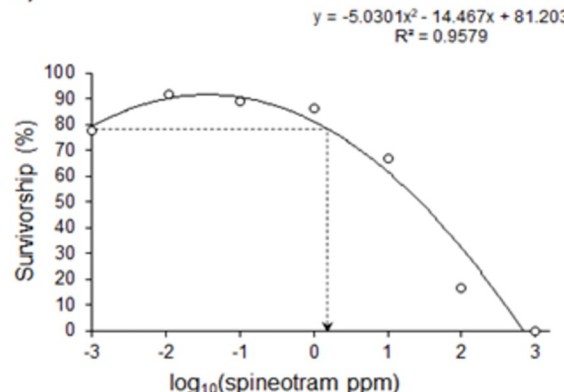

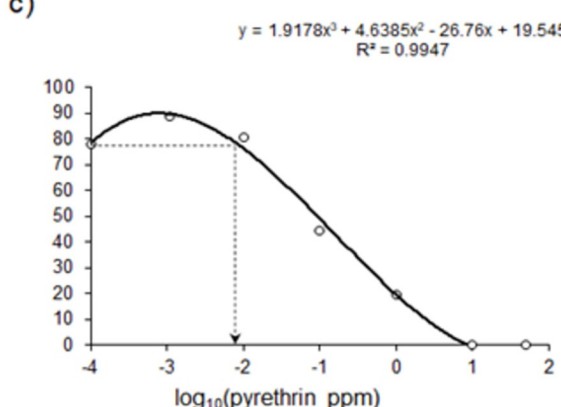

**Fig 1. Initial dose-response curves.** Initial dose-response curves and equations for zeta-cypermethrin (a), spinetoram (b), and pyrethrin (c) with the dotted lines representing the zero equivalent point. Log transformations for zeta-cypermethrin and spinetoram were log10(ppm + 0.001) and log10(ppm + 0.0001) for pyrethrin.

**Table 2. Concentrations tested for each insecticide treatment and their corresponding lethal concentrations.** Confidence intervals (95%) are shown in parentheses.

| Treatment | zeta-cypermethrin (ppm) | spinetoram (ppm) | pyrethrin (ppm) |
|---|---|---|---|
| $LC_0$ | 0 | 0 | 0 |
| $LC_{10}$ | 0.005 (0.001–0.039) | 0.05 (0.10–0.227) | 0.001 (0.0001–0.004) |
| $LC_{20}$ | 0.050 (0.009–0.391) | 0.35 (0.073–1.637) | 0.005 (0.001–0.017) |
| $LC_{30}$ | 0.300 (0.045–2.056) | 1.43 (0.302–6.804) | 0.014 (0.004–0.048) |
| $LC_{40}$ | 1.20 (0.186–8.497) | 4.84 (1.019–22.989) | 0.033 (0.009–0.115) |

**Table 3. Three-way ANOVA results for the effects of fly sex (male or female), treatment concentration (LC$_0$, LC$_{10}$, LC$_{20}$, LC$_{30}$, LC$_{40}$) and type of exposure (single or double) on survival for each insecticide experiment.** Bolded values indicate statistically significant p-values (P ≤ 0.05).

| Factor | zeta-cypermethrin | | spinetoram | | pyrethrin | |
|---|---|---|---|---|---|---|
| sex | $F_{1,180} = 15.90$ | **P<0.0001** | $F_{1,180} = 10.67$ | **P<0.001** | $F_{1,180} = 15.33$ | **P<0.0001** |
| treatment | $F_{4,180} = 5.38$ | **P<0.0001** | $F_{4,180} = 9.55$ | **P<0.0001** | $F_{4,180} = 5.03$ | **P<0.001** |
| exposure | - | - | $F_{1,180} = 1.01$ | P = 0.314 | $F_{1,180} = 1.19$ | P = 0.277 |
| sex* treatment | $F_{4,180} = 0.70$ | P = 0.596 | $F_{4,180} = 2.23$ | P = 0.068 | $F_{4,180} = 5.22$ | **P<0.001** |
| sex*exposure | - | - | $F_{1,180} = 0.053$ | P = 0.818 | $F_{1,180} = 3.58$ | P = 0.060 |
| treatment*exposure | - | - | $F_{4,180} = 1.88$ | P = 0.116 | $F_{4,180} = 1.43$ | P = 0.225 |
| sex*treatment*exposure | - | - | $F_{4,180} = 0.23$ | P = 0.924 | $F_{4,180} = 0.49$ | P = 0.744 |

Single- and double-exposure were only tested for the spinetoram and pyrethrin experiments, hence the exposure factors are not included in the model for zeta-cypermethrin.

controls and the higher concentration treatments but less so for the LC$_{10}$ and LC$_{20}$ treatments. Table 4 shows that sub-lethal exposure to zeta-cypermethrin had no effect on pupal rate (Fig 3A), on the eclosion success of F1 flies (Fig 4A), or on the adult mass of F1 flies (Fig 5A).

Despite having the highest initial LC$_{50}$ of all the insecticides, the spinetoram treatment had the lowest survivorship across the sub-lethal concentrations. Survivorship in the controls only averaged ~40% for females and 13% for males. A significant effect of sex and treatment was found, as well as a marginally significant sex*treatment interaction, but no effect of exposure type (Table 3). Female flies had significantly higher survivorship than males, but the overall trends were similar. Fig 2B and 2C shows that survivorship was highest in the LC$_0$ and LC$_{10}$ concentrations for males and in the LC$_{10}$ and LC$_{20}$ treatments for females. Again, however, the LC$_0$, LC$_{10}$, and LC$_{20}$ concentrations were statistically similar, with a significant decline in survivorship occurring at the LC$_{30}$ and LC$_{40}$ concentrations for both sexes (S1 Table). S1B and S1E Fig shows that survivorship sharply declined around day 6 for the LC$_{40}$ treatment but not until day 9 for the other treatments. Sub-lethal concentrations of spinetoram did not impact pupal rate (Fig 3B) but did significantly impact eclosion success. Fig 4B shows that eclosion success was highest for the LC$_{10}$ treatment and significantly lower for the LC$_{40}$ treatment (S2 Table). There was a significant interaction between treatment concentration and exposure type on F1 fly mass (Table 4). Average adult mass did not differ across concentrations in the single-exposure group; however, adult mass was significantly higher in the LC$_{30}$ and LC$_{40}$ treatments for the chronic exposure group (S4 Table, Fig 5B).

For pyrethrin, survivorship was high across all treatments and there was a significant interaction between sex and treatment (Table 3). Fig 2D and 2E shows that female survivorship was comparable across all concentrations in both the single- and double-exposure treatments. Male survivorship was high and comparable across the 0 to LC$_{30}$ treatments but significantly lower in the LC$_{40}$ treatment for both single- and double-exposures (Fig 2D and 2E). Survivorship was statistically similar for males and females in the LC$_0$-LC$_{30}$ treatments but males exhibited significantly lower survivorship than females in the LC$_{40}$ treatments (S4 Table), which S1C and S1F Fig shows was due to a decline in male survivorship around day 5. There was a significant effect of treatment and exposure type on pupal rate (Table 4, Fig 3C) but no other significant effects on F1 eclosion success (Fig 4C) or adult mass (Fig 5C). Interestingly, pupal rate was highest in the LC$_{30}$ treatment for both single- and double-exposure treatments and higher overall in the double-exposure treatments (S5 Table).

Across all insecticides, which can only be compared qualitatively, pupal rate was similar in the zeta-cypermethrin and spinetoram experiments but considerably higher in the pyrethrin

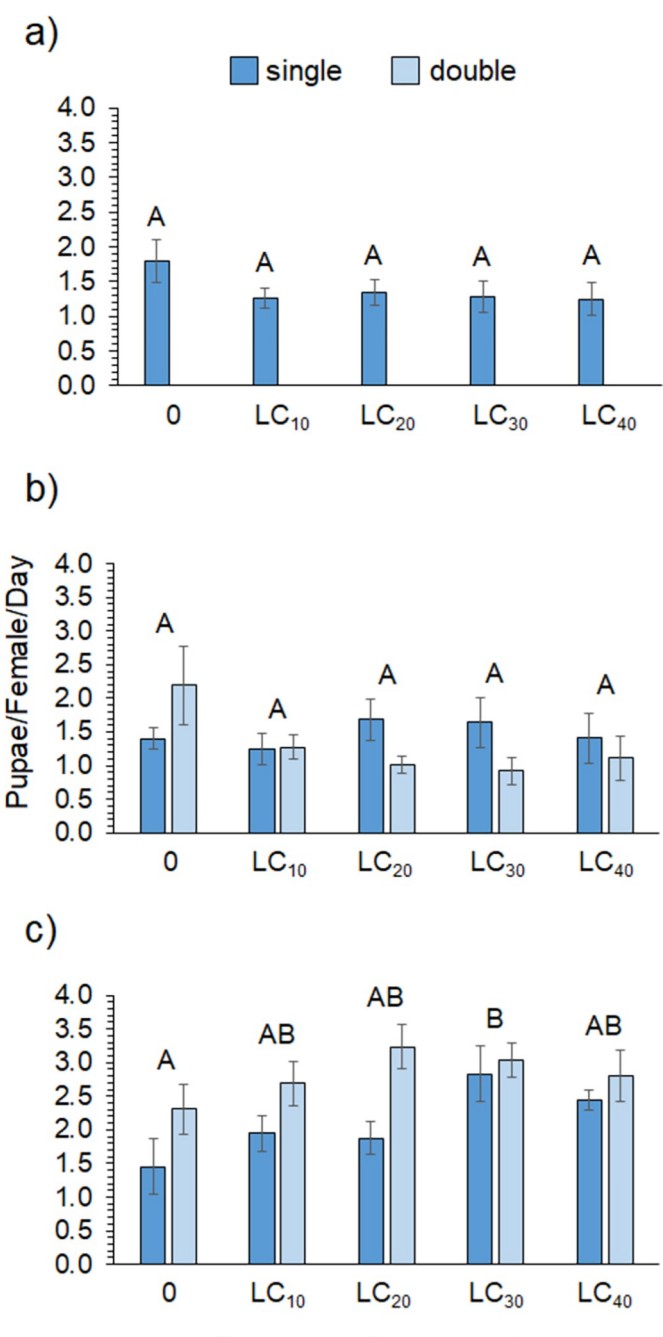

**Fig 2. Survivorship.** Average survivorship for male and female flies across sub-lethal concentrations of (a) zeta-cypermethrin, (b) single-exposure and (c) double-exposure to spinetoram, (d) single-exposure and (d) double-exposure to pyrethrin (N = 10). Different letters indicate significant post-hoc differences between treatments or sex where significant main effects or interactions were found (S1 and S2 Tables; Tukey's Test, P ≤ 0.05).

experiment (Fig 3A–3C). Eclosion success of F1 flies was also similar for the zeta-cypermethrin and spinetoram insecticides but considerably lower in the pyrethrin treatments (Fig 4A–4C), ranging from 7.5–41.5% compared to 24.0–44.7% for zeta-cypermethrin and 21.8–45.8% for spinetoram. Adult mass of F1 flies was similar across insecticides (Fig 5A–5C).

**Table 4. Three-way ANOVA results for the effects of treatment concentration ($LC_0$, $LC_{10}$, $LC_{20}$, $LC_{30}$, $LC_{40}$) and type of exposure (single or double) on female reproduction, including pupal rate (pupae rate/female), the eclosion success of F1 offspring, and the average mass of F1 offspring for each insecticide experiment.** Bolded values indicate statistically significant p-values ($P \leq 0.05$).

| Variable | Factor | zeta-cypermethrin | | spinetoram | | pyrethrin | |
|---|---|---|---|---|---|---|---|
| Pupal Rate | treatment | $F_{4,45} = 0.46$ | $P = 0.761$ | $F_{4,89} = 0.96$ | $P = 0.435$ | $F_{4,90} = 2.87$ | **P = 0.027** |
| | exposure | - | - | $F_{1,89} = 2.38$ | $P = 0.128$ | $F_{1,90} = 11.61$ | **P = 0.001** |
| | treatment*exposure | - | - | $F_{4,89} = 1.33$ | $P = 0.266$ | $F_{4,90} = 0.97$ | $P = 0.427$ |
| F1 Eclosion Success | treatment | $F_{4,45} = 2.16$ | $P = 0.089$ | $F_{4,85} = 3.33$ | **P = 0.014** | $F_{4,83} = 0.42$ | $P = 0.796$ |
| | exposure | - | - | $F_{1,85} = 2.77$ | $P = 0.100$ | $F_{1,83} = 3.13$ | $P = 0.081$ |
| | treatment*exposure | - | - | $F_{4,85} = 0.61$ | $P = 0.659$ | $F_{4,83} = 2.01$ | $P = 0.100$ |
| F1 Adult Mass | treatment | $F_{4,45} = 0.62$ | $P = 0.650$ | $F_{4,79} = 1.54$ | $P = 0.198$ | $F_{4,83} = 0.28$ | $P = 0.893$ |
| | exposure | - | - | $F_{1,79} = 0.67$ | $P = 0.416$ | $F_{1,83} = 1.76$ | $P = 0.188$ |
| | treatment*exposure | - | - | $F_{4,79} = 5.33$ | **P = 0.001** | $F_{4,83} = 0.65$ | $P = 0.626$ |

Single- and double-exposure were only tested for the spinetoram and pyrethrin experiments, hence the exposure factors is not included in the model for zeta-cypermethrin.

### Transgenerational effects

For all insecticides, and in corroboration with the sub-lethal data, males had higher overall mortality than female flies (Fig 6A–6C). Table 5 shows that a significant parental treatment*sex interaction was found for pyrethrin, with Fig 4C indicating that F1 male morality was only higher in the $LC_{15}$ and $LC_{20}$ parental treatments but similar across the other treatments. Overall, mortality was considerably lower in the pyrethrin experiment (Fig 6C), despite using a higher $LC_{50}$ diagnostic dose.

Overall, results showed comparable mortality with a weak trend of slightly higher mortality for males and slightly lower mortality for females in some sub-lethal parental treatments. For instance, Fig 6A shows that female mortality was lower than the controls in the zeta-cypermethrin $LC_{10}$ and $LC_{25}$ treatments. Fig 6B also shows lower female mortality than the controls in the spinetoram $LC_{15}$ treatment. For pyrethrin, Fig 6C, mortality is lower than controls in the $LC_{10}$ and $LC_{30}$ treatments for males and in the $LC_{15}$ and $LC_{20}$ treatments for females (S6 Table). However, despite showing a significant effect of sex on F1 survivorship for the zeta-cypermethrin and spinetoram treatments, there was no significant effect of parental exposure for either sex (Table 4).

### Discussion

Overall, we found strong sex-specific effects of sub-lethal concentrations on spotted-wing Drosophila survival (Table 3). In general, male mortality was higher than female mortality, even across control treatments, and this pattern was evident in both the hormesis and transgenerational experiments. Sub-lethal exposure had largely similar effects on survival across insecticides and showed a trend of hormesis, as survivorship was the same or higher than the control treatments in at least one sub-lethal dose for each insecticide (Fig 2A–2E). For instance, survivorship was higher than the control in the $LC_{10}$ and $LC_{20}$ treatments for males and the $LC_{20}$ treatment for females in the zeta-cypermethrin experiment (Fig 2A). A similar pattern was evident for spinetoram (Fig 2B and 2C) and pyrethrin (Fig 2D and 2E). Despite this trend, the differences between the controls and other concentrations were not statistically significant. These results, however, advocate further testing, as it is possible that the number of doses we tested limited our ability to detect a maximized hormetic response. It is also possible that our sample size limited our statistical power to detect differences [106]. After assessing over 11,000 dose-

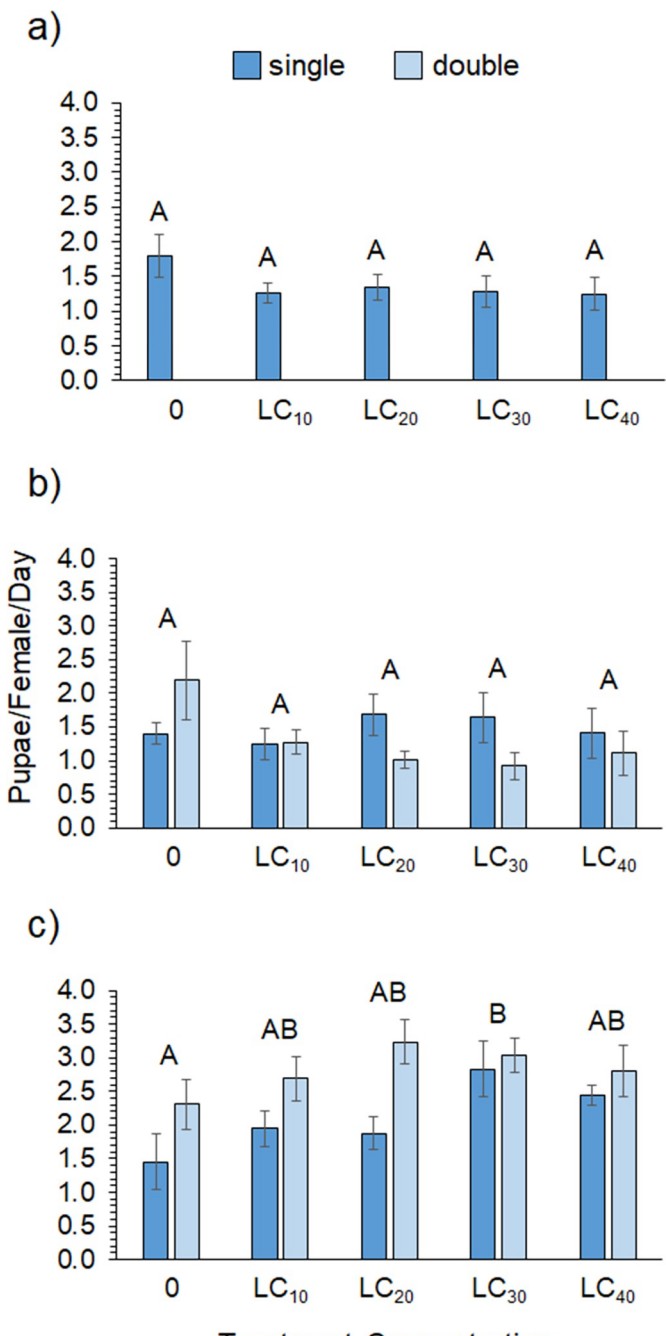

**Fig 3. Pupation success.** Pupal rate across single- and double-exposures for (a) zeta-cypermethrin (N = 19–20), (b) spinetoram (N = 9–10), and (c) pyrethrin (N = 10). The last collection of pupae from the pyrethrin treatment were too numerous to count accurately, so data only include the first two collections. Different letters indicate significant post-hoc differences between treatments (Tukey's Test, P ≤ 0.05).

responses in animal, microbe, and plant systems, [107] showed that the maximum hormetic stimulation increased with the number of doses assessed below the zero equivalent point (ZEP), i.e., the dose where the response is the same as the control group. This indicates that the number of doses tested and the dose range surveyed have a significant impact on the ability

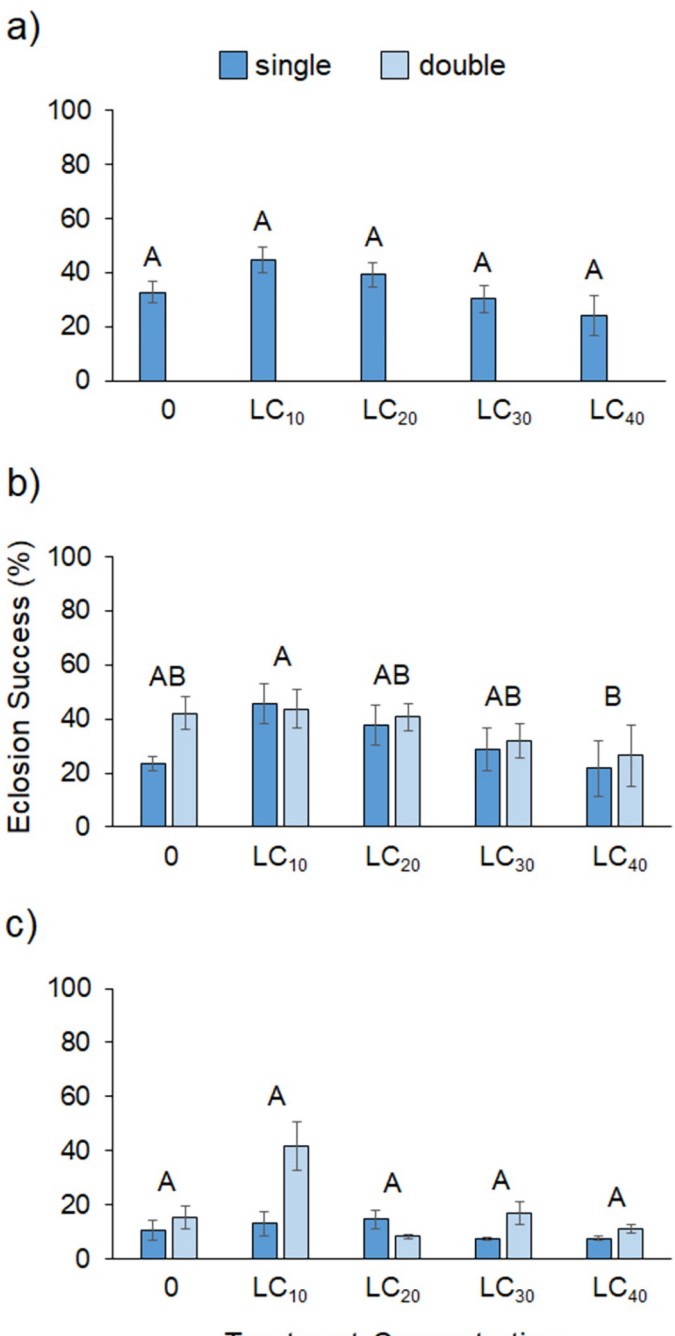

**Fig 4. Eclosion success.** Eclosion success of offspring across single- and double-exposures for (a) zeta-cypermethrin (N = 17–20), (b) spinetoram (N = 8–10), and (c) pyrethrin (N = 5–10). Different letters indicate significant post-hoc differences between treatments (Tukey's Test, P ≤ 0.05).

to detect hormesis and quantify its magnitude of stimulation. As shown in Fig 1A–1C, the ZEP for the initial dose-response curves for zeta-cypermethrin, spinetoram, and pyrethrin were 0.05, 1.67, 0.0085 respectively, resulting in only one dose for zeta-cypermethrin, three doses for spinetoram, and two doses for pyrethrin being tested below the ZEP. Given that [107] suggest testing six doses below the ZEP, it is likely that the doses tested were too restrictive to elicit

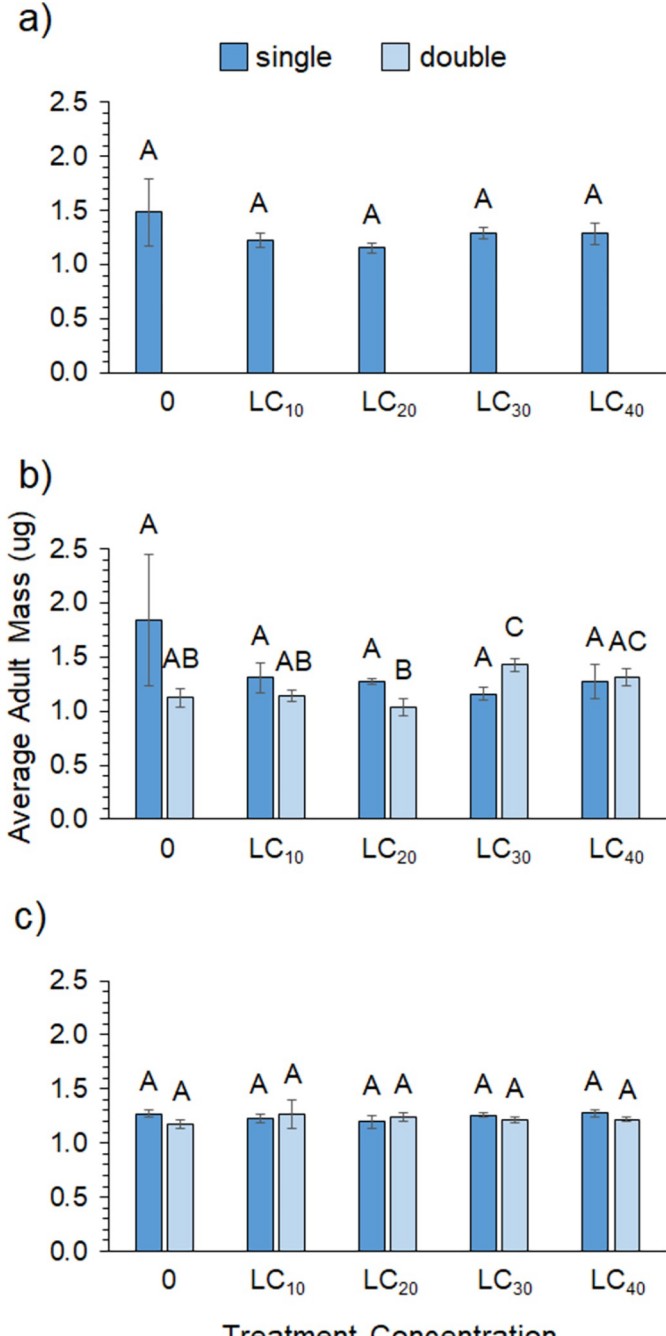

**Fig 5. Adult mass.** Adult mass of offspring across single- and double-exposures for (a) zeta-cypermethrin (N = 11–20), (b) spinetoram (N = 8–10), and (c) pyrethrin (N = 5–10). Different letters indicate significant post-hoc differences between each exposure group across treatments (Tukey's Test, P ≤ 0.05).

the maximum hormetic stimulation for all insecticides. Despite this, we tested a comparable or greater number of doses than several other studies that found evidence for insecticide hormesis [81, 85, 88]. Ultimately, higher-resolution studies will be needed to better determine if these insecticides are likely to produce hormetic effects in spotted-wing Drosophila and what those dose ranges would be.

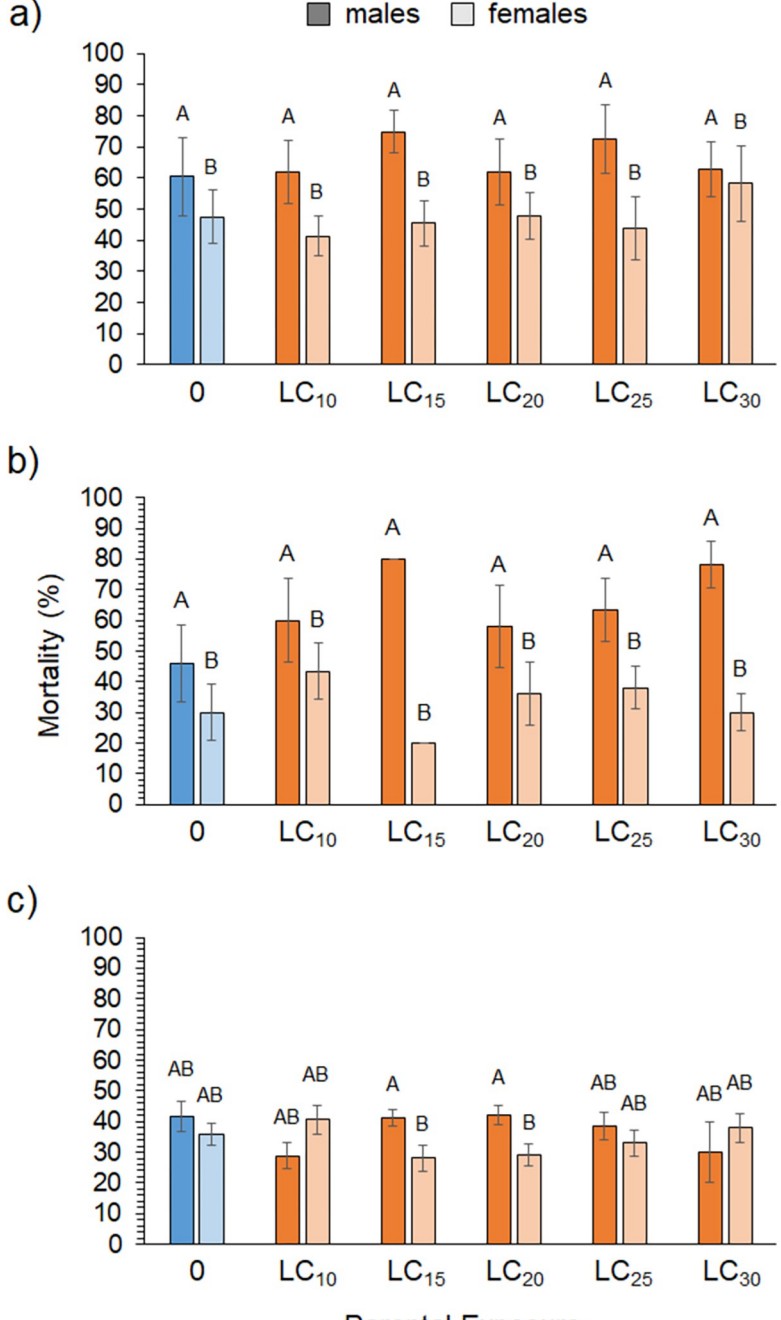

**Fig 6. F1 Mortality.** Average mortality of male (darker bars) and female (lighter bars) F1 flies from (a) zeta-cypermethrin (N = 6–10), (b) spinetoram (N = 1–10), and (c) pyrethrin (N = 6–10) parental exposure treatments after exposure to a diagnostic dose. Mortality of parental flies exposed to no insecticides are shown in blue, all other exposure concentrations are shown in orange. Different letters indicate significant post-hoc differences between the sexes for each treatment (Tukey's Test, $P \leq 0.05$).

For female reproductive performance the data also exhibited some evidence of hormesis. The eclosion success data exhibited an inverted-U shape curve for all insecticides, but differences between controls and other concentrations were only evident for spinetoram (Fig 4B). There were three specific instances where hormetic effects were statistically supported by the

**Table 5. Two-way ANOVA results for the impacts of sex and parental exposure to zeta-cypermethrin, spinetoram, or pyrethrin on adult F1 mortality after exposure to a diagnostic dose.** Bolded values indicate statistically significant p-values (P-value ≤ 0.05).

| Factor | zeta-cypermethrin | spinetoram | pyrethrin |
|---|---|---|---|
| parental treatment | $F_{5,96} = 0.265$, P = 0.931 | $F_{5,105} = 0.740$, P = 0.595 | $F_{5,94} = 0.931$, P = 0.918 |
| sex | $F_{1,96} = 11.36$, **P = 0.001** | $F_{1,105} = 18.60$, **P<0.001** | $F_{1,94} = 0.1222$, P = 0.272 |
| parental treatment*sex | $F_{5,96} = 0.545$, P = 0.742 | $F_{5,105} = 0.664$, P = 0.651 | $F_{5,94} = 2.676$, **P = 0.026** |

performance data. Pyrethrin pupal rates were higher overall for the double- versus the single-exposure treatments. Also, the $LC_{30}$ treatments for both exposure groups showed significantly higher pupal rates than the controls (Fig 3C). For spinetoram, average adult mass was also significantly higher in the $LC_{30}$ treatment than the controls but only for the chronic treatment (Fig 5B). These results show that some spotted-wing Drosophila performance variables do respond hormetically to sub-lethal exposure and suggest that female reproductive performance may be stimulated at low dose ranges. Of course, in order to understand the effect that these responses may have on fly population dynamics, more work will be required to determine their ultimate impact on overall fecundity and offspring viability.

An interesting result from hormesis experiments was that the mortality observed across the sub-lethal treatments was somewhat discordant from what was expected based on the initial dose-response bioassays for each insecticide. For example, zeta-cypermethrin survivorship was slightly lower than expected across all sub-lethal doses, while pyrethrin survivorship was much higher than expected, exhibiting ~90% survival across all sub-lethal treatments (Fig 2D and 2E). Spinetoram survivorship was markedly lower than expected, with 100% mortality observed in the $LC_{40}$ treatment (Fig 2C and 2D). These results are particularly surprising given that the initial dose-response assays were performed only a month before the sub-lethal experiments, limiting the likelihood that the populations used for the initial dose-responses and the sub-lethal experiments were genetically different due to lab selection or drift. Overall, these results suggest that either an external unknown factor may have impacted the lab colony's susceptibility to spinetoram or that the colony contained a large amount of genetic variability across genes related to insecticide resistance and detoxification. The genetic variability hypothesis is bolstered by the fact that the discrepancy between the initial dose-response curves and experimental results were similar for the insecticides with the same mode of action. Zeta-cypermethrin and pyrethrin are both sodium channel modulators [108, 109], while spinetoram is a spinosad that acts on GABA-gated chloride channels [110, 111]. The sex-specific differences in survivorship, on the other hand, seem to be impacted by the intensity of the insecticide, as differences between male and female flies were greater in the zeta-cypermethrin and spinetoram treatments, both synthetic insecticides, than the pyrethrin treatment, an organic insecticide. However, this likely a reflection of the fact that pyrethrin had more limited effects overall on mortality than the other two insecticides, at least at the sub-lethal doses tested (Fig 2). The higher survivorship exhibited by female versus male flies has important implications for population dynamics and pest management, particularly for spotted-wing Drosophila in fruit production systems where oviposition in high-value crops is the primary driver of reductions is marketable yields.

While the transgenerational results did show a trend of higher mortality in male F1 flies from parents exposed to zeta-cypermethrin and spinetoram, differences between the controls and treatments were not statistically different (Fig 6). These data do not indicate that parental exposure had any positive impact on offspring susceptibility; however, we only subjected F1 flies to one diagnostic dose. Without a full dose-response assay, which was not possible in this study due to the limited number of F1 flies available, we cannot determine whether

susceptibility at other doses was affected. It is also unknown how parental exposure may have impacted other fitness characteristics in F1 flies, such as performance variables. Several other studies that have looked at and detected transgenerational effects exposed parental populations for multiple generations [81, 112, 113], which may be necessary for hormetic traits to exhibit in offspring. To better assess the potential for transgenerational effects to play a role in spotted-wing Drosophila population dynamics, future research should focus on multiple fitness characteristics, including survival and performance, measured over a broader range of insecticide concentrations administered in acute and chronic exposures.

In conclusion, the results of this study show variable effects of sub-lethal insecticide exposure on spotted-wing Drosophila survival, with potential hormetic effects on survival and demonstrable impacts on female reproductive traits. Additionally, the methodological uncertainties discussed above highlight the difficulties that exist in studying hormetic effects. Unlike the stabilizing responses to lethal insecticide exposure, i.e., the inevitability of 100% mortality, hormetic and transgenerational effects represent transient maxima occurring at some point along an insect's dose-response curve; a point that is often unknown *a priori* and can be easily missed if the dose range and/or number of concentrations tested do not provide adequate resolution. This requires testing a large number of treatments with high replication and makes identifying these responses logistically challenging. Despite this, it is important to account for the impact that these lesser-studied responses have on pest population dynamics, particularly in agricultural systems. While the impact that sub-lethal effects have on insecticide efficacy may be indirect, ambiguous, and seemingly minor, their cumulative effects are potentially significant and widespread [37, 40, 114, 115].

There are many factors that can reduce effective exposure in the field, including effective spraying procedure, the structural complexity of the host, insecticide degradation due to environmental factors, and the movement of insects in-and-out, as well as, within the system. Depending on the level of sub-lethal exposure, results can have negative, neutral, or positive impacts on insect fitness, which can directly affect pest population dynamics but can also have indirect effects. For example, sub-lethal exposure, by definition, doesn't lead to instant mortality, and therefore, doesn't directly contribute to the evolution of genetic resistance through natural selection. However, it can produce resistant phenotypes intra-generationally through hormesis and inter-generationally via epigenetic effects. Poor nutrition [116–118], exposure to various environmental stressors [119–123] and insecticides have been shown to produce transgenerational tolerance in many different taxa, including insects. Given that epigenetic changes can be instantaneous and widespread, transgenerational effects have the capacity to have significantly impacts on insecticide efficacy in agricultural systems. However, because epigenetic marks are reversible, transgenerational effects can be short lived in the absence insecticide use [124], manifesting in short outbreaks, or they can have longer-term effects on the development of resistance traits [40, 124]. The creation and perpetuation of phenotypic variability can impact natural selection by serving as adaptive intermediate stages [40, 101–103]. Yet, the relative contribution of these responses to the development of insecticide resistance is not well understood. In fact, even the evolution of genetic resistance in the field is still quite mysterious, as resistance can readily develop to multiple unique chemistries within the same species but is often lost at rates faster than predicted by natural selection [40, 125–127]. Genetic bottlenecks also do not appear to limit the evolution of resistance, as would be expected [40, 128, 129]. In light of this, researchers have hypothesized that plastic responses to insecticides may play an important role, either through direct impacts on mutation rates or via epigenetic effects [40, 128, 129]. In any case, much more research is needed to better understand this role, as well as the more generalized impacts of sub-lethal exposure on insecticide viability, particularly for pests that are difficult to control, such as spotted-wing Drosophila.

## Supporting information

**S1 Fig. Time-dependent mortality.** Survival curves for male (a-c) and female (d-f) flies from the (a, d) zeta-cypermethrin, (b, e) spinetoram, and (c, f) pyrethrin experiments (N = 10). (TIF)

**S1 Table. Survivorship post-hoc results for treatment effects.** Tukey's post-hoc test P-values for the significant treatment effects on survival from the three-way ANOVA. Bolded values indicate statistically significant P-values (P-value ≤ 0.05). (PDF)

**S2 Table. Eclosion success post-hoc results for spinetoram.** Tukey's post-hoc test P-values for the significant treatment effects on eclosion success for the spinetoram treatment. Bolded values indicate statistically significant P-values (P-value ≤ 0.05). (PDF)

**S3 Table. Treatment*exposure post-hoc results for spinetoram.** Tukey's post-hoc test P-values for the significant treatment*exposure interaction on adult mass for the spinetoram treatment. Bolded values indicate statistically significant P-values (P-value ≤ 0.05). (PDF)

**S4 Table. Survivorship post-hoc results for pyrethrin.** Tukey's post-hoc test results for the significant sex*treatment interaction on lifespan for the pyrethrin treatment. Bolded values indicate statistically significant P-values (P-value ≤ 0.05). (PDF)

**S5 Table. Pupal rate post-hoc results for pyrethrin.** Tukey's post-hoc test P-values for the significant treatment effects on pupal rate for the pyrethrin treatment. Bolded values indicate statistically significant P-values (P-value ≤ 0.05). (PDF)

**S6 Table. Treatment*sex interaction post-hoc results for F1 mortality.** Tukey's post-hoc test P-values for the significant parental treatment*sex interaction on F1 mortality for the pyrethrin treatment. Bolded values indicate statistically significant P-values (P-value ≤ 0.05). (PDF)

## Acknowledgments

We would like to thank Eric Burkness and Dominique Ebbenga for their assistance with fly colony maintenance and input on experimental design.

## Author Contributions

**Conceptualization:** Carrie Deans, William D. Hutchison.

**Data curation:** Carrie Deans.

**Formal analysis:** Carrie Deans.

**Methodology:** Carrie Deans.

**Supervision:** William D. Hutchison.

**Writing – original draft:** Carrie Deans.

**Writing – review & editing:** Carrie Deans, William D. Hutchison.

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
