## [Decision Letter · Decision Letter 0]

31 May 2022

PONE-D-22-09651Hormetic and Transgenerational Effects in Spotted-wing Drosophila (Diptera: Drosophilidae) in Response to Three Commonly-Used InsecticidesPLOS ONE

Dear Dr. Deans,

Thank you for submitting your manuscript to PLOS ONE. After careful consideration, we feel that it has merit but does not fully meet PLOS ONE’s publication criteria as it currently stands. Therefore, we invite you to submit a revised version of the manuscript that addresses the points raised during the review process.

We look forward to receiving your revised manuscript.

Kind regards,

Giancarlo López-Martínez, Ph.D.

Academic Editor

PLOS ONE

Journal Requirements:

“This work was supported by funding from the Agricultural Growth, Research, and Innovation (AGRI) Crops Research Program, of the Minnesota Department of Agriculture, St. Paul, MN (2018-2021), and the Minnesota Agricultural Experiment Station, University of Minnesota, St. Paul, MN, USA. Grants were awarded to WH.”           

“This work was supported by 420 funding from the Agricultural Growth, Research, and Innovation (AGRI) Crops Research 421 Program, of the Minnesota Department of Agriculture, St. Paul, MN (2018-2021), and 422 the Minnesota Agricultural Experiment Station, University of Minnesota, St. Paul, MN, 423 USA. Grants were awarded to WH”

“This work was supported by funding from the Agricultural Growth, Research, and Innovation (AGRI) Crops Research Program, of the Minnesota Department of Agriculture, St. Paul, MN (2018-2021), and the Minnesota Agricultural Experiment Station, University of Minnesota, St. Paul, MN, USA. Grants were awarded to WH.”

Additional Editor Comments:

The reviewers have identified a few areas where the story and the manuscript can be improved with some very minor corrections. As most of the points brought up by the reviewers will serve to improve the current version of the MS, I invite the authors to take some time in addressing these comments and I look forward to an updated version.

Thank you

Reviewers' comments:

Reviewer's Responses to Questions

**Comments to the Author**

1. Is the manuscript technically sound, and do the data support the conclusions?

Reviewer #1: Yes

Reviewer #2: Partly

2. Has the statistical analysis been performed appropriately and rigorously? 

Reviewer #1: Yes

Reviewer #2: Yes

3. Have the authors made all data underlying the findings in their manuscript fully available?

Reviewer #1: No

Reviewer #2: Yes

4. Is the manuscript presented in an intelligible fashion and written in standard English?

Reviewer #1: Yes

Reviewer #2: Yes

5. Review Comments to the Author

Reviewer #1: General comments

This is a good paper. Good hypotheses, good methods, good interpretation, although things get unwieldy in places (too much into the weeds on minor results that distract from key findings). I have offered some minor edits and a few papers that may or may not be of interest related to the topics discussed.

Specific comments

1. L80. It is usually advisable to avoid the word “beneficial” and instead use the word “stimulatory” when referring to hormetic effects. Beneficial effects of one endpoint measure can only be assessed in the context of other effects over the long-term, which are rarely measured in published paper. The word “stimulatory” will be more precise and accurate in most instances.

2. L95-99. Yes, under-explored but potential impacts in agriculture are becoming more appreciated, e.g. Sci Total Environ, 2022, 825, 153899

3. L100-112. Links of insecticide induced hormesis with transgenerational effects, epigenetics, resistance, and enzyme induction have been studied, e.g. papers on hormesis by Rix, Ayyanath, Cutler

4. L162-164. Modify. You are not calculating exact concentrations. You are using the data to generate a linear model that estimates LC values, some of which you hypothesize will induce stress responses reflective of hormesis

5. L170. You are exposing insects to prescribed concentrations, not doses

6. L171. Suggest using different terminology. Acute vs chronic is not differentiated one exposure vs two exposures, but rather short vs long-term exposure. Just call this what it is, being more precise and accurate: one exposure vs two exposures.

7. L171. Why was the two-time exposure not done with zeta-cypermethrin?

8. L176. Clarify. In the one-time exposure, flies were exposed for 4 h, and thereafter unexposed to insecticide for the remainder of their lives. In two-time exposure, flies were immediately exposed for 4 h, then unexposed for 116 h following, and then exposed for another 4 hours – correct? Were the “new vials” untreated with insecticide.

9. L183. Parental (P) generation adult flies?

10. L186. New untreated vials?

11. L191-192. Why do you say approximate LC25 or LC50 DD concentration. Had you not previously determined ‘exact’ estimates of LC25 and LC50 values?

12. L193. … pyrethrin).

13. L219. More complete results of the probit analyses should be presented: Chi-square, confidence intervals, slopes, df, etc. These data may be informative in interpreting hormesis results, e.g. lower slope is indicative of a more heterogeneous response to the insecticide, which in turn might correspond to a more pronounced hormetic response in the population, particularly across generations where interactive tolerance/resistance factors might be more likely to come into play. Even as is table 1 lack proper headings and is presented bass-ackwards. (‘Insecticide’ should be the left-most column, thereafter with sequential presentation of lethal concentrations)

14. L243. The very low control survival in spinetoram treatments relative to the other insecticides is striking. So low that I wonder if these data should even be included. I assume this is discussed later in the Discussion.

15. L311. Section 3.1 of this paper talks about some the experimental considerations for detecting hormesis. Perhaps more important in the number of doses/concentrations is replication within treatments to reduce type 2 error: Calabrese EJ. 2005. Paradigm lost, paradigm found…. Environ Poll 138, 378–411

16. L351-353. Seems a more likely and parsimonious explanation is that something unknown and external to the fly population, in and of itself, affected the fly population. It is doubtful genetic variability in a lab colony is so great to account for this. I bet if you did the experiment again today you’d get very different (better) survival for spinetoram treatments.

17. L391. Yes. I suspect hormetic responses, though subtle in many (not all) cases, are ubiquitous in the field and are simply not realized due to lack of study.

18. L412. These papers may be of interest as it relates to hormesis, tolerance to insecticides, resistance, etc. Pest management science 74 (2), 314-322; Journal of pest science 89 (2), 581-589; Science of The Total Environment 827, 154085

Reviewer #2: Hormetic and Transgenerational Effects in Spotted-wing Drosophila (Diptera: Drosophilidae) in Response to Three Commonly-Used Insecticides

Dear Authors:

I am glad to see more researchers examining hormesis in the context of agriculture. I have provided some specific comments to assist in editing the manuscript. By way of summary, there are areas of the introduction, methods, results, and discussion that need to be amended. The authors spend time in the introduction discussing epigenetic mechanisms of hormesis, however this is not an aspect of their paper, therefore this should be edited to be proportionate with their work and proposed potential future work. There are a number of areas in methods where not enough detail is included, and I have provided specific comments on these areas. There are a number of errors in describing the results. The results from the probit analysis are missing. This should be included. There is some confusion with mortality and survival, which confounds the findings. In addition, the figures need to be edited for consistency, and colour scheme changed. The discussion is heavily lacking in comparison and discussion of other works examining insecticide induced hormesis in insects. This is surprising as this field has grown considerably in the last 5-10 years. I have provided the authors with a number of examples of papers in this field that may be pertinent to their work, which also have works cited in this field, and I highly suggest the authors search the literature for more work on insecticide induced hormesis. Despite this, I think the authors are on solid footing with their work, and following edits, this paper should be ready for publication.

Review Comments

The abstract is well constructed and clear. I like that the authors began by discussing factors that can alter the effective exposure of pests to insecticides. This is key to understanding the legitimate implications of hormesis in agricultural systems.

Keywords

I would suggest the authors replace epigenetics with transgenerational effects, where epigenetics is often more associated with the examination of heritable effects on the genome not involving changes in nucleotide sequences, such as gene expression, gene methylation, histone modification etc, rather than strictly phenotypic changes. This could be misleading as to what was specifically studied, if interpreted as studying effects on the molecular level.

Introduction

Line 57: I think here it would be better to start this as follows:

Spotted-wing Drosophila (Drosophila suzukii Matsumura), an invasive species native to Asia, has become a serious pest in fruit-producing areas throughout the United States and Europe, following its establishment in the early 2000s.

Line 61: What is the purpose of saying “certain climates”? It is a bit too vague to provide pertinent information. Be a bit more specific, or do not include at all.

Line 63: You might want to include some specifics on regions and specific crops where D. suzukii outbreaks have been major issues.

Line 71:

“This can occur by impeding contact between the insecticide and spotted-wing Drosophila adults, especially considering that spotted-wing Drosophila are small, highly mobile, and crepuscular insects that have been documented to exhibit daily movement patterns in 74 several crops.”

Be careful of long or run-on sentences. The writing of this manuscript is very good, but when sentences are too long it confounds their meaning. I would suggest reading back through the manuscript sentence by sentence, and look for areas where you can break up your sentences, or adjust your sentences.

Line 84: “Hormetic and transgenerational effects can not only increase plant damage through

direct reductions in insecticide efficacy, they may also promote the evolution of

insecticide resistance through their impacts on phenotypic diversity”

Be careful of wording here. Any increases in plant damage that would be associated directly with hormesis would be due to stimulatory effects induced on the insect, such as stimulation in fecundity/fertility (thus increased feeding on the plant), stimulation of longevity (thus longer lifetime of feeding), stimulation of feeding (thus more feeding), etc. The reduced efficacy of the insecticide is presumably already in existence if the insect is undergoing hormesis as hormesis occurs in concentration ranges generally below the NOAEL. Hormesis and transgenerational effects in this context are a function of reduced insecticide efficacy, hormesis does not act through reducing insecticide efficacy, which is how your statement is worded.

With regard to information on hormesis and pesticide resistance and impacts on phenotypic diversity or plasticity, see works by Costantini. Some examples:

Costantini D. Hormesis Promotes Evolutionary Change. Dose Response 2019; 17: 1559325819843376.

Costantini D. Does hormesis foster organism resistance to extreme events? Frontiers in Ecology and the Environment 2014; 12: 209-210.

Lines 96: “It is particularly surprising that the effects of hormesis have not been more thoroughly addressed in agricultural pest management, considering that there are a multitude of factors in agricultural settings that may reduce effective exposure and there are significant costs associated with insecticide failures due to increases in insect tolerance or genetic resistance.”

This has been examined and discussed by authors such as Cutler, Guedes, Rix, Ullah, amongst some others. Transgenerational effects are also addressed in these. For some examples see:

Cutler, G.C., Amichot, M., Benelli, G., Guedes, R.N.C., Qu, Y., Rix, R.R., Ullah, F., Desneux, N. (2022). Hormesis and insects: Effects and interactions in agroecosystems. Science of the Total Environment. doi: 10.1016/j.scitotenv.2022.153899

Ullah F, Gul HN, Desneux N, Gao XW, Song DL. Imidacloprid-induced hormesis effects on demographic traits of the melon aphid, Aphis gossypii. Entomologia Generalis 2019; 39: 325-337.

Ullah F, Gul H, Tariq K, Desneux N, Gao XW, Song DL. Thiamethoxam induces transgenerational hormesis effects and alteration of genes expression in Aphis gossypii. Pesticide Biochemistry and Physiology 2020; 165: 11.

Rix, R.R., Cutler, G.C. (2018). Does multigenerational exposure to hormetic concentrations of imidacloprid precondition aphids for increased insecticide tolerance? Pest Management Science. 74: 314-322.

Cutler GC, Guedes RNC. Occurrence and Significance of Insecticide-Induced Hormesis in Insects. Pesticide Dose: Effects on the Environment and Target and Non-Target Organisms. 1249. American Chemical Society, 2017, pp. 101-119.

Rix, R.R., Ayyanath, M.M., Cutler, G.C. (2016). Sublethal concentrations of imidacloprid increase reproduction, alter expression of detoxification genes, and prime Myzus persicae for subsequent stress. Journal of Pest Science. 89: 581-589.

Guedes RN, Cutler GC. Insecticide-induced hormesis and arthropod pest management. Pest Manag Sci 2014; 70: 690-7.

Ayyanath MM, Cutler GC, Scott-Dupree CD, Sibley PK. Transgenerational shifts in reproduction hormesis in green peach aphid exposed to low concentrations of imidacloprid. PLoS One 2013; 8: e74532

Cutler GC. Insects, insecticides and hormesis: evidence and considerations for study. Dose Response 2013; 11: 154-77.

Line 101: Transgenerational effects can also refer to hormetic stimulatory effects (such as increased fecundity) occurring across generations, not just increased tolerance in offspring. The papers cited above also have examples of this.

Lines 100-122: These paragraphs include some excellent information on potential epigenetic mechanisms of hormesis. As mentioned above regarding the abstract, the authors are not examining epigentic mehcnaisms, thus the information might be more suited to a small area of the discussion or conclusion as something that might need to be focused on in the future. Mentioning it at the outset of the paper and in a large section is misleading the reader into thinking it is part of your experimental focus. Your focus should be on the phenotypic effects observed from other studies. Given the fact the authors have missed a number of key studies/authors in this field, I would suggest perhaps searching through the agricultural hormesis literature further and focus there, rather than brining in the extraneous epigenetic work.

Methods

Fly Culture

Can the authors provide more details on the fly rearing? Were they kept in containers? Approximately how many flies per container? What are the specific details of the diet? Etc. My suggestion would be to describe it in as much detail as you would if you were giving someone an SOP (Standard Operating Procedure).

Insecticide Exposure

Make sure you include any necessary trademarks and company/manufacturer names and locations when using the brand names of the insecticides. Examples for how this may look can be found in the publications suggested in previous comments.

Dose-response assays

Can the authors list the concentrations used for each insecticide?

Hormesis protocol

The authors refer to their two-time exposure as a chronic exposure. I would suggest just calling it two-time exposure. Chronic exposure can more often be interpreted as being a constant exposure or constantly reoccurring exposure. I would not consider a two-time exposure to be chronic.

Transgenerational protocol

Were the F1 adults of the same age or close in age when exposed to the diagnostic/discriminating dose?

Also, some punctuation is missing throughout the paragraph. The copy editor may pick up on this as well.

Results

Dose-response assays

The authors should include the statistics associated with a probit analysis, ex: chi-square, fiducial limits for endpoints of interest (LC50, hormetic LC, etc).

Hormetic effects

I think the authors are mixing up mortality and survival here. Fig 2a shows that the LC10 and LC20 treatments had the highest survival for males and females, not the highest mortality as stated in the text. And survivorship decreases significantly at the LC30 and LC40 concentrations in Fig 2a. It does not increase as was stated in the text.

I would also suggest the authors include the supplementary material as main material.

Line 243: “Despite having the highest initial LC50 of all the insecticides, the spinetoram treatment had the lowest survivorship across the sub-lethal concentrations. Survivorship in the controls only averaged ~40% for females and 13% for males.”

Given the large control mortality (40%) for females, it may not actually be that the spinetoram has the lowest survivorship, it may simply be a weak group of insects. There seems to be a vast difference between survivorship in controls with experiments corresponding to Figs 2b,c. From close to 80-100% survival to less than 10% in some cases, although it does seem proportional to the treatments. The authors should discuss and try to account for this.

Transgenerational effects

Line 287: LC15 should be LC10

Line 290: “Results showed a trend toward comparable or lower mortality for some sub-lethal parental treatments”. While there are some very small trends, if anything, there appears to be slightly increased mortality in most concentrations compared with controls. However, there are no statistically significant differences. Line 291: “For instance, Fig 6a shows that male mortality was lower than controls in the zeta-cypermethrin LC10 and LC25 treatments for female flies.” This is not what I see in this figure, although the trends mentioned in the other figures are visible. The legend is also in grey, while the bars in the figure are in orange, which may be confounding the authors findings.

Discussion

Line 310: Despite this trend, the differences between the controls and other concentrations were not statistically significant. These results, however, advocate further testing, as it is possible that the number of doses we tested limited our ability to detect a maximized hormetic response.

Yes, indeed your results warrant further testing, you did see some small trends. I would agree that your concentration ranges may have limited your ability to detect the maximized hormetic dose. The range in which the maximal hormetic responses occur can be wide ranging, thus it can be difficult to detect in a first experiment. You may want to consider next time testing a number of concentrations below the LC10 and even LC1. Calabrese indicates that frequently the hormetic concentration falls below the NOAEL, however, with insects, studies have shown that hormetic effects can occur outside the NOAEL. Looking at transgenerational effects as well could certainly impact where the hormetic concentration could fall. A concentration that is not hormetic in one generation, could in fact be hormetic in another generation (as you saw with your eclosion data).

I like the fact the authors include a lot of analysis of what they found, unwinding their findings for the reader. I would suggest the authors find more works on hormesis in insects in the literature (there are lots) and weave this into the discussion, to perhaps assist their insights. For example, the authors could discuss other works on transgenerational effects, the ranges of concentrations other studies used compared with theirs. This would be a nice addition to the authors’ existing discussion on concentration ranges. The discussion is highly lacking in examples of hormesis in insects in response to insecticides, and the connection to agriculture. There is solid work on this. I have provided a number of examples, however there are many others, and the authors should provide some analysis of their work in relation to other studies.

Tables and Figures:

Table 4: Represent the degrees of freedom with the F-value as per the previous tables.

I would suggest the authors use greyscale for their figures, rather than colour. If they want to use colour, I would suggest using darker colours. These colours are not aesthetically pleasing.

The authors should be consistent with their formatting. For example, Fig 1 has the y-axis survivorship label on all the individual graphs, whereas the other figures do not. The graphs in Fig 6 are also boxed in. The top and right side lines should be removed from each graph to be consistent with the other graphs. Only the bottom x-axis, and left y-axis are necessary for these graphs.

6. PLOS authors have the option to publish the peer review history of their article (what does this mean?). If published, this will include your full peer review and any attached files.

Reviewer #1: No

Reviewer #2: No

---

## [Author Response · Author response to Decision Letter 0]

17 Jun 2022

Editor:

Our response to specific Reviewer comments, are noted below (Italics):

Reviewer #1: General comments

This is a good paper. Good hypotheses, good methods, good interpretation, although things get unwieldy in places (too much into the weeds on minor results that distract from key findings). I have offered some minor edits and a few papers that may or may not be of interest related to the topics discussed.

Specific comments

1. L80. It is usually advisable to avoid the word “beneficial” and instead use the word “stimulatory” when referring to hormetic effects. Beneficial effects of one endpoint measure can only be assessed in the context of other effects over the long-term, which are rarely measured in published paper. The word “stimulatory” will be more precise and accurate in most instances.

I don’t see the term “beneficial” used here, but perhaps you are referring to the use of “negative” and “positive” effects. We have changed these terms to “inhibitory” and “stimulatory”.

2. L95-99. Yes, under-explored but potential impacts in agriculture are becoming more appreciated, e.g. Sci Total Environ, 2022, 825, 153899

The Cutler et al., 2022 (ref 51) was added here.

3. L100-112. Links of insecticide induced hormesis with transgenerational effects, epigenetics, resistance, and enzyme induction have been studied, e.g. papers on hormesis by Rix, Ayyanath, Cutler

Three more citations were added on line 103 (Ayyanath et al., 2013; Rix et al., 2015; Ullah et al., 2020).

4. L162-164. Modify. You are not calculating exact concentrations. You are using the data to generate a linear model that estimates LC values, some of which you hypothesize will induce stress responses reflective of hormesis

The wording was changed to emphasize that the concentrations were generated from a model and used to estimate dose-mortality relationships.

5. L170. You are exposing insects to prescribed concentrations, not doses

The word “dose” was removed.

6. L171. Suggest using different terminology. Acute vs chronic is not differentiated one exposure vs two exposures, but rather short vs long-term exposure. Just call this what it is, being more precise and accurate: one exposure vs two exposures.

We changed the treatment names to “single-exposure” and “double-exposure” regimes rather than acute and chronic (line 174-177).

7. L171. Why was the two-time exposure not done with zeta-cypermethrin?

We made the decision to add another treatment after we had already completed the zeta-cypermethrin trials.

8. L176. Clarify. In the one-time exposure, flies were exposed for 4 h, and thereafter unexposed to insecticide for the remainder of their lives. In two-time exposure, flies were immediately exposed for 4 h, then unexposed for 116 h following, and then exposed for another 4 hours – correct? Were the “new vials” untreated with insecticide.

Yes, after the 4-hr exposure period the flies were moved into new vials containing no insecticide and remained unexposed for the rest of their lifespan. The double-exposure flies were treated for 4-hrs, then moved into clean vials and treated again in the same manner 5 days later. We changed the wording in this section to make these details clearer (line 179-182).

9. L183. Parental (P) generation adult flies?

We specified that parental treatments will be indicated by P and made changes throughout the document.

10. L186. New untreated vials?

We added the term “untreated” (line191).

11. L191-192. Why do you say approximate LC25 or LC50 DD concentration. Had you not previously determined ‘exact’ estimates of LC25 and LC50 values?

12. L193. … pyrethrin).

End parenthesis was added.

13. L219. More complete results of the probit analyses should be presented: Chi-square, confidence intervals, slopes, df, etc. These data may be informative in interpreting hormesis results, e.g. lower slope is indicative of a more heterogeneous response to the insecticide, which in turn might correspond to a more pronounced hormetic response in the population, particularly across generations where interactive tolerance/resistance factors might be more likely to come into play. Even as is table 1 lack proper headings and is presented bass-ackwards. (‘Insecticide’ should be the left-most column, thereafter with sequential presentation of lethal concentrations)

Table 1 was intended to show the concentrations that we used for each sub-lethal treatment and their justification based on their associated mortalities but we did add another table that shows more information about the initial dose-response bioassays we performed (line 223-225). Table 1 show the information regarding the dose-response curves and Table 2 now shows the treatment concentrations. 

14. L243. The very low control survival in spinetoram treatments relative to the other insecticides is striking. So low that I wonder if these data should even be included. I assume this is discussed later in the Discussion.

We do discuss this result in the Discussion (line 354-361), although we cannot explain really it. While it is a surprising result, it is a consistent result across the sexes and the single- and double-exposure treatments so we will leave it in the manuscript but discuss its anomalous nature. 

15. L311. Section 3.1 of this paper talks about some the experimental considerations for detecting hormesis. Perhaps more important in the number of doses/concentrations is replication within treatments to reduce type 2 error: Calabrese EJ. 2005. Paradigm lost, paradigm found…. Environ Poll 138, 378–411

We do briefly discuss this point on line 320 but have added this reference there.

16. L351-353. Seems a more likely and parsimonious explanation is that something unknown and external to the fly population, in and of itself, affected the fly population. It is doubtful genetic variability in a lab colony is so great to account for this. I bet if you did the experiment again today you’d get very different (better) survival for spinetoram treatments.

True. We included a sentence about external impacts (line360).

17. L391. Yes. I suspect hormetic responses, though subtle in many (not all) cases, are ubiquitous in the field and are simply not realized due to lack of study.

18. L412. These papers may be of interest as it relates to hormesis, tolerance to insecticides, resistance, etc. Pest management science 74 (2), 314-322; Journal of pest science 89 (2), 581-589; Science of The Total Environment 827, 154085

Yes, thank you. These references were added on line 423.

Reviewer #2: Hormetic and Transgenerational Effects in Spotted-wing Drosophila (Diptera: Drosophilidae) in Response to Three Commonly-Used Insecticides

Dear Authors:

I am glad to see more researchers examining hormesis in the context of agriculture. I have provided some specific comments to assist in editing the manuscript. By way of summary, there are areas of the introduction, methods, results, and discussion that need to be amended. The authors spend time in the introduction discussing epigenetic mechanisms of hormesis, however this is not an aspect of their paper, therefore this should be edited to be proportionate with their work and proposed potential future work. There are a number of areas in methods where not enough detail is included, and I have provided specific comments on these areas. There are a number of errors in describing the results. The results from the probit analysis are missing. This should be included. There is some confusion with mortality and survival, which confounds the findings. In addition, the figures need to be edited for consistency, and colour scheme changed. The discussion is heavily lacking in comparison and discussion of other works examining insecticide induced hormesis in insects. This is surprising as this field has grown considerably in the last 5-10 years. I have provided the authors with a number of examples of papers in this field that may be pertinent to their work, which also have works cited in this field, and I highly suggest the authors search the literature for more work on insecticide induced hormesis. Despite this, I think the authors are on solid footing with their work, and following edits, this paper should be ready for publication.

Review Comments

The abstract is well constructed and clear. I like that the authors began by discussing factors that can alter the effective exposure of pests to insecticides. This is key to understanding the legitimate implications of hormesis in agricultural systems.

Keywords

I would suggest the authors replace epigenetics with transgenerational effects, where epigenetics is often more associated with the examination of heritable effects on the genome not involving changes in nucleotide sequences, such as gene expression, gene methylation, histone modification etc, rather than strictly phenotypic changes. This could be misleading as to what was specifically studied, if interpreted as studying effects on the molecular level.

Respectfully, we feel that transgenerational effects appropriately fall under the broader field of epigenetics, in that they are mediated by the epigenetic mechanisms described above. Although this study does not directly address the molecular aspects of epigenetics, it is important to us that transgenerational effects are acknowledged as epigenetic phenomenon because it highlights the ability of responses that are not strictly genetically-determined to play a role in resistance, which is an underappreciated point.

Introduction

Line 57: I think here it would be better to start this as follows:

Spotted-wing Drosophila (Drosophila suzukii Matsumura), an invasive species native to Asia, has become a serious pest in fruit-producing areas throughout the United States and Europe, following its establishment in the early 2000s.

Ok, revised; however, as the authority name was placed in parentheses, as recommended by the ESA (Entomol. Soc. of America). 

Line 61: What is the purpose of saying “certain climates”? It is a bit too vague to provide pertinent information. Be a bit more specific, or do not include at all.

We were referring to climates where winter temperatures do not allow for overwintering but have removed the phrase to improve clarity.

Line 63: You might want to include some specifics on regions and specific crops where D. suzukii outbreaks have been major issues.

It's difficult to pinpoint different spots because spotted-wing has become established in every U.S. state except Arizona, and despite this, data relating to specific economic costs is still rather limited (e.g., CA, and MN). As noted in Asplen et al. (2015), (and Refs # 3-5), SWD has become a major pest in nearly every U.S. state and EU country, where preferred host crops are available (raspberry, strawberry, cherry, etc.). With such low tolerances for fruit damage, rather low to moderate populations can have economic impacts (Refs #10-12). Because of this we feel it is more productive to discuss the areas where resistance is becoming a problem rather than its general pest status, which we discuss on line 70.

Line 71:

“This can occur by impeding contact between the insecticide and spotted-wing Drosophila adults, especially considering that spotted-wing Drosophila are small, highly mobile, and crepuscular insects that have been documented to exhibit daily movement patterns in 74 several crops.”

Be careful of long or run-on sentences. The writing of this manuscript is very good, but when sentences are too long it confounds their meaning. I would suggest reading back through the manuscript sentence by sentence, and look for areas where you can break up your sentences, or adjust your sentences.

We simplified the sentence and scanned for other run-on sentences.

Line 84: “Hormetic and transgenerational effects can not only increase plant damage through direct reductions in insecticide efficacy, they may also promote the evolution of insecticide resistance through their impacts on phenotypic diversity”

Be careful of wording here. Any increases in plant damage that would be associated directly with hormesis would be due to stimulatory effects induced on the insect, such as stimulation in fecundity/fertility (thus increased feeding on the plant), stimulation of longevity (thus longer lifetime of feeding), stimulation of feeding (thus more feeding), etc. The reduced efficacy of the insecticide is presumably already in existence if the insect is undergoing hormesis as hormesis occurs in concentration ranges generally below the NOAEL. Hormesis and transgenerational effects in this context are a function of reduced insecticide efficacy, hormesis does not act through reducing insecticide efficacy, which is how your statement is worded.

Good point. We reworded the sentence.

With regard to information on hormesis and pesticide resistance and impacts on phenotypic diversity or plasticity, see works by Costantini. Some examples:

Costantini D. Hormesis Promotes Evolutionary Change. Dose Response 2019; 17: 1559325819843376.

Costantini D. Does hormesis foster organism resistance to extreme events? Frontiers in Ecology and the Environment 2014; 12: 209-210.

Thanks. References were added.

Lines 96: “It is particularly surprising that the effects of hormesis have not been more thoroughly addressed in agricultural pest management, considering that there are a multitude of factors in agricultural settings that may reduce effective exposure and there are significant costs associated with insecticide failures due to increases in insect tolerance or genetic resistance.”

This has been examined and discussed by authors such as Cutler, Guedes, Rix, Ullah, amongst some others. Transgenerational effects are also addressed in these. For some examples see:

Cutler, G.C., Amichot, M., Benelli, G., Guedes, R.N.C., Qu, Y., Rix, R.R., Ullah, F., Desneux, N. (2022). Hormesis and insects: Effects and interactions in agroecosystems. Science of the Total Environment. doi: 10.1016/j.scitotenv.2022.153899

Ullah F, Gul HN, Desneux N, Gao XW, Song DL. Imidacloprid-induced hormesis effects on demographic traits of the melon aphid, Aphis gossypii. Entomologia Generalis 2019; 39: 325-337.

Ullah F, Gul H, Tariq K, Desneux N, Gao XW, Song DL. Thiamethoxam induces transgenerational hormesis effects and alteration of genes expression in Aphis gossypii. Pesticide Biochemistry and Physiology 2020; 165: 11.

Rix, R.R., Cutler, G.C. (2018). Does multigenerational exposure to hormetic concentrations of imidacloprid precondition aphids for increased insecticide tolerance? Pest Management Science. 74: 314-322.

Cutler GC, Guedes RNC. Occurrence and Significance of Insecticide-Induced Hormesis in Insects. Pesticide Dose: Effects on the Environment and Target and Non-Target Organisms. 1249. American Chemical Society, 2017, pp. 101-119.

Rix, R.R., Ayyanath, M.M., Cutler, G.C. (2016). Sublethal concentrations of imidacloprid increase reproduction, alter expression of detoxification genes, and prime Myzus persicae for subsequent stress. Journal of Pest Science. 89: 581-589.

Guedes RN, Cutler GC. Insecticide-induced hormesis and arthropod pest management. Pest Manag Sci 2014; 70: 690-7.

Ayyanath MM, Cutler GC, Scott-Dupree CD, Sibley PK. Transgenerational shifts in reproduction hormesis in green peach aphid exposed to low concentrations of imidacloprid. PLoS One 2013; 8: e74532

Cutler GC. Insects, insecticides and hormesis: evidence and considerations for study. Dose Response 2013; 11: 154-77.

Thanks. We have added all of these references and cited them on line 97.

Line 101: Transgenerational effects can also refer to hormetic stimulatory effects (such as increased fecundity) occurring across generations, not just increased tolerance in offspring. The papers cited above also have examples of this.

While other researchers may categorize increased fecundity and other hormetic effects on reproduction as transgenerational effects, we feel that this can be misleading, given that these responses occur within the parent, not the offspring. It is important to make the distinction between processes occurring in the parent due to the exposure of the parent and those occurring in the offspring due to the exposure of the parent but not the offspring themselves (or the exposure of the offspring at early developmental stages, i.e., egg exposure). This is an important point because it specifies the mechanisms involved, which often get convoluted based on semantical differences in definitions. In our view, transgenerational effects are reserved for situations where expressional changes in the offspring are caused by parental exposure to a stimulus and the absence of direct exposure of the offspring. These responses implicate epigenetic processes, as they persist across meiosis/mitosis.

Lines 100-122: These paragraphs include some excellent information on potential epigenetic mechanisms of hormesis. As mentioned above regarding the abstract, the authors are not examining epigentic mehcnaisms, thus the information might be more suited to a small area of the discussion or conclusion as something that might need to be focused on in the future. Mentioning it at the outset of the paper and in a large section is misleading the reader into thinking it is part of your experimental focus. Your focus should be on the phenotypic effects observed from other studies. Given the fact the authors have missed a number of key studies/authors in this field, I would suggest perhaps searching through the agricultural hormesis literature further and focus there, rather than brining in the extraneous epigenetic work.

Ok. We moved a large portion of this paragraph to the Discussion and included more specific content about transgenerational effects reported in agricultural systems in the Introduction (line 107).

Methods

Fly Culture

Can the authors provide more details on the fly rearing? Were they kept in containers? Approximately how many flies per container? What are the specific details of the diet? Etc. My suggestion would be to describe it in as much detail as you would if you were giving someone an SOP (Standard Operating Procedure).

We added a more detailed description on line 146.

Insecticide Exposure

Make sure you include any necessary trademarks and company/manufacturer names and locations when using the brand names of the insecticides. Examples for how this may look can be found in the publications suggested in previous comments.

This information was added on line 157.

Dose-response assays

Can the authors list the concentrations used for each insecticide?

This information was added on line 167.

Hormesis protocol

The authors refer to their two-time exposure as a chronic exposure. I would suggest just calling it two-time exposure. Chronic exposure can more often be interpreted as being a constant exposure or constantly reoccurring exposure. I would not consider a two-time exposure to be chronic.

Noted. We changed the terminology to single- and double-exposure for these treatments.

Transgenerational protocol

Were the F1 adults of the same age or close in age when exposed to the diagnostic/discriminating dose?

Yes, we added the term “newly-eclosed” to indicate this on line 201.

Also, some punctuation is missing throughout the paragraph. The copy editor may pick up on this as well.

We fixed the punctuation issues.

Results

Dose-response assays

The authors should include the statistics associated with a probit analysis, ex: chi-square, fiducial limits for endpoints of interest (LC50, hormetic LC, etc).

Yes, we added a new table (Table 1) that contains this information and a brief discussion about the probit analysis on line 229.

Hormetic effects

I think the authors are mixing up mortality and survival here. Fig 2a shows that the LC10 and LC20 treatments had the highest survival for males and females, not the highest mortality as stated in the text. And survivorship decreases significantly at the LC30 and LC40 concentrations in Fig 2a. It does not increase as was stated in the text.

We corrected the results to indicate that survivorship was highest, not mortality.

I would also suggest the authors include the supplementary material as main material.

Respectfully, we feel that with 5 tables and 6 figures already in the main document, the best place for the remaining tables and figures are in the supplemental information section, particularly because most of the information contained in them is denoted on the figures (posthoc letters) or discussed in text (time course of mortality).

Line 243: “Despite having the highest initial LC50 of all the insecticides, the spinetoram treatment had the lowest survivorship across the sub-lethal concentrations. Survivorship in the controls only averaged ~40% for females and 13% for males.”

Given the large control mortality (40%) for females, it may not actually be that the spinetoram has the lowest survivorship, it may simply be a weak group of insects. There seems to be a vast difference between survivorship in controls with experiments corresponding to Figs 2b,c. From close to 80-100% survival to less than 10% in some cases, although it does seem proportional to the treatments. The authors should discuss and try to account for this.

All the insecticide trials were done at the same time with the same group of flies, so it is not likely that the insects were weaker. We do discuss this anomalous result in the Discussion with potential explanations (line 360).

Transgenerational effects

Line 287: LC15 should be LC10

The posthoc letters in Fig 4c indicate that male mortality was higher than females in the LC15 and LC20 treatments.

Line 290: “Results showed a trend toward comparable or lower mortality for some sub-lethal parental treatments”. While there are some very small trends, if anything, there appears to be slightly increased mortality in most concentrations compared with controls. However, there are no statistically significant differences. 

Correct. Upon reassessment, the text was changed to better reflect the general trends.

Line 291: “For instance, Fig 6a shows that male mortality was lower than controls in the zeta-cypermethrin LC10 and LC25 treatments for female flies.” This is not what I see in this figure, although the trends mentioned in the other figures are visible. The legend is also in grey, while the bars in the figure are in orange, which may be confounding the authors findings.

The grey legend is to show that the male are in a darker shade than the females and the figure does show that female mortality was slighter lower than the female control values for the LC10 and LC25 treatments.

Discussion

Line 310: Despite this trend, the differences between the controls and other concentrations were not statistically significant. These results, however, advocate further testing, as it is possible that the number of doses we tested limited our ability to detect a maximized hormetic response.

Yes, indeed your results warrant further testing, you did see some small trends. I would agree that your concentration ranges may have limited your ability to detect the maximized hormetic dose. The range in which the maximal hormetic responses occur can be wide ranging, thus it can be difficult to detect in a first experiment. You may want to consider next time testing a number of concentrations below the LC10 and even LC1. Calabrese indicates that frequently the hormetic concentration falls below the NOAEL, however, with insects, studies have shown that hormetic effects can occur outside the NOAEL. Looking at transgenerational effects as well could certainly impact where the hormetic concentration could fall. A concentration that is not hormetic in one generation, could in fact be hormetic in another generation (as you saw with your eclosion data).

I like the fact the authors include a lot of analysis of what they found, unwinding their findings for the reader. I would suggest the authors find more works on hormesis in insects in the literature (there are lots) and weave this into the discussion, to perhaps assist their insights. For example, the authors could discuss other works on transgenerational effects, the ranges of concentrations other studies used compared with theirs. This would be a nice addition to the authors’ existing discussion on concentration ranges. The discussion is highly lacking in examples of hormesis in insects in response to insecticides, and the connection to agriculture. There is solid work on this. I have provided a number of examples, however there are many others, and the authors should provide some analysis of their work in relation to other studies.

Given the current length of our Discussion, and several new references already added, we do not feel that we have space to add many more examples from the literature. However, we have added a short comparison of our concentration range with other studies (line 337) and the methodology of other transgenerational studies (line 394). We believe additional examples, beyond what we now provide, is beyond the scope of this paper. Perhaps a future paper, such as a Review article, may be needed to fully update this topic for other researchers? 

Tables and Figures:

Table 4: Represent the degrees of freedom with the F-value as per the previous tables.

Table 5 was edited to match Table 4.

I would suggest the authors use greyscale for their figures, rather than colour. If they want to use colour, I would suggest using darker colours. These colours are not aesthetically pleasing.

We decided to keep the colors but changed them to the darker colors used on Fig6 to maintain consistency.

The authors should be consistent with their formatting. For example, Fig 1 has the y-axis survivorship label on all the individual graphs, whereas the other figures do not. The graphs in Fig 6 are also boxed in. The top and right side lines should be removed from each graph to be consistent with the other graphs. Only the bottom x-axis, and left y-axis are necessary for these graphs.

We fixed the axes and removed the lines around Fig 6 to maintain consistency.

Thanks again for the excellent questions and suggestions for improvement of the manuscript.

---

## [Editor Report · Decision Letter 1]

30 Jun 2022

Hormetic and transgenerational effects in spotted-wing Drosophila (Diptera: Drosophilidae) in response to three commonly-used insecticides

PONE-D-22-09651R1

Dear Dr. Deans,

We’re pleased to inform you that your manuscript has been judged scientifically suitable for publication and will be formally accepted for publication once it meets all outstanding technical requirements.

Kind regards,

Giancarlo López-Martínez, Ph.D.

Academic Editor

PLOS ONE

Additional Editor Comments (optional):

Dear authors:

I really appreciate the detailed effort put forth to address the comments brought by the reviewers. In my opinion, the MS was solid during the initial submission, but many of the changes made make aspects of this neat story clearer to the wider audience of this journal. Personally, and as a hormesis/transgenerational person, these results are really cool, and I really appreciate a thorough rebuttal letter.

Thank you,

GC
---

## [Editor Report · Acceptance letter]

12 Jul 2022

PONE-D-22-09651R1 

Hormetic and transgenerational effects in spotted-wing Drosophila (Diptera: Drosophilidae) in response to three commonly-used insecticides 

Dear Dr. Deans:

I'm pleased to inform you that your manuscript has been deemed suitable for publication in PLOS ONE. Congratulations! Your manuscript is now with our production department. 

Kind regards, 

on behalf of

Dr. Giancarlo López-Martínez 

Academic Editor

PLOS ONE